# Activation of an Effective Immune Response after Yellow Fever Vaccination Is Associated with the Genetic Background and Early Response of IFN-γ and CLEC5A

**DOI:** 10.3390/v13010096

**Published:** 2021-01-12

**Authors:** Tamiris Azamor, Andréa Marques Vieira da Silva, Juliana Gil Melgaço, Ana Paula dos Santos, Caroline Xavier-Carvalho, Lucia Elena Alvarado-Arnez, Leonardo Ribeiro Batista-Silva, Denise Cristina de Souza Matos, Camilla Bayma, Sotiris Missailidis, Ana Paula Dinis Ano Bom, Milton Ozorio Moraes, Patrícia Cristina da Costa Neves

**Affiliations:** 1Bio-Manguinhos/Fundação Oswaldo Cruz, Rio de Janeiro 21040-900, Brazil; tamiris.azamor@bio.fiocruz.br (T.A.); amarques@bio.fiocruz.br (A.M.V.d.S.); juliana.melgaco@bio.fiocruz.br (J.G.M.); apaula@bio.fiocruz.br (A.P.d.S.); dmatos@bio.fiocruz.br (D.C.d.S.M.); camilla.bayma@bio.fiocruz.br (C.B.); sotiris.missailidis@bio.fiocruz.br (S.M.); adinis@bio.fiocruz.br (A.P.D.A.B.); 2Instituto Oswaldo Cruz, Fundação Oswaldo Cruz, Rio de Janeiro 21040-900, Brazil; caroline_xc@yahoo.com.br (C.X.-C.); milton.moraes@fiocruz.br (M.O.M.); 3National Research Coordination, Universidad Privada Franz Tamayo, Cochabamba 4780, Bolivia; lucia.alvarado@unifranz.edu.bo; 4Molecular Carcinogenesis Program—Research Coordination, Instituto Nacional do Câncer, Rio de Janeiro 20231-050, Brazil; leoribio@hotmail.com

**Keywords:** yellow fever vaccine, interferon gamma, CLEC5A

## Abstract

The yellow fever vaccine (YF17DD) is highly effective with a single injection conferring protection for at least 10 years. The YF17DD induces polyvalent responses, with a TH1/TH2 CD4^+^ profile, robust T CD8^+^ responses, and synthesis of interferon-gamma (IFN-γ), culminating in high titers of neutralizing antibodies. Furthermore, C-type lectin domain containing 5A (CLEC5A) has been implicated in innate outcomes in other flaviviral infections. Here, we conducted a follow-up study in volunteers immunized with YF17DD, investigating the humoral response, cellular phenotypes, gene expression, and single nucleotide polymorphisms (SNPs) of IFNG and CLEC5A, to clarify the role of these factors in early response after vaccination. Activation of CLEC5A^+^ monocytes occurred five days after vaccination (DAV). Following, seven DAV data showed activation of CD4^+^ and CD8^+^T cells together with early positive correlations between type II IFN and genes of innate antiviral response (STAT1, STAT2, IRF7, IRF9, OAS1, and RNASEL) as well as antibody levels. Furthermore, individuals with genotypes rs2430561 AT/AA, rs2069718 AG/AA (IFNG), and rs13237944 AC/AA (CLEC5A), exhibited higher expression of IFNG and CLEC5A, respectively. Together, we demonstrated that early IFN-γ and CLEC5A responses, associated with rs2430561, rs2069718, and rs13237944 genotypes, may be key mechanisms in the long-lasting immunity elicited by YF17DD.

## 1. Introduction

Yellow fever (YF) is a potentially fatal mosquito-borne disease caused by the neurovirulent and viscerotropic YF virus (YFV), from the genus flavivirus, family *Flaviviridae*. The mainstay of YF control involves control of the mosquito vectors *Aedes Albopictus* and *Aedes aegypti*, and vaccination [1]. The attenuated YF vaccine was developed in 1937 by Theiler and Smith through serial passages of the YFV wild-type Asibi strain in mice and chick minced embryo culture, generating the non-neurovirulent strain 17D which protected monkeys from challenge with a virulent strain of the virus with production of neutralizing antibodies [2]. After independent subcultures in chick embryo tissue, it was generated the strains of 17DD (YF17DD) used as a vaccine in South America, and 17D-204, used in the rest of the world [3,4]. Genome sequencing studies revealed mutations in non-coding regions, and amino acid differences in structural and non-structural proteins, including non-conservative variations in E299, E305, E331 and E380 within ectodomain III of envelope protein, the cell binding region [5,6,7,8]. Further, it was shown that YF vaccines present much lower quasispecies diversity, when compared with virulent strains [9,10]. Thence, the YF vaccine has been considered a gold standard vaccine, with an excellent efficacy and safety record, in that a single full dose is sufficient to confer sustained immunity and potentially life-long protection against YFV [11,12,13]. Clinical trials using one-fifth fractionated dose revealed that 17D confers protection for at least 10 years [14,15]. Despite its great efficacy, the cellular and molecular mechanisms by which the vaccine virus induces such protection are not yet fully elucidated [4].

The attenuated YFV is injected subcutaneously, mimetizing the natural route of entrance by mosquito bites, where the virus infects and activates human dendritic cells (DCs). As a response, the activated DCs express type I and III interferons and different pattern recognition receptors (toll-like receptors-2,7,8 and 9), and migrate to the lymph nodes, presenting virus antigens to a diverse repertoire of T cells [16,17,18]. Hence, the early events after YF vaccination consist of (i) a complex modulation of innate immune cytokines, (ii) a polyvalent cellular response, with a mixed proinflammatory TH1/regulatory TH2 CD4^+^ cell profile, which results in a robust CD8^+^ T cell response, and a complex modulation of innate immune cytokines, culminating with (iii) production of neutralizing antibodies, which are the primary correlate of protection [19]. In this way, early events following immunization have a key role in determining the strength and quality of the adaptive immune response that develops. It was demonstrated that the primary immunization with the vaccine results in early release of the type II interferon, interferon-gamma (IFN-γ), a cytokine which improved the acquired immune responses in mice, and the levels of neutralizing antibodies in humans [20,21]. A growing body of evidence indicates that the production of IFN-γ is regulated by single nucleotide polymorphisms (SNPs) within the IFNG gene, which may influence host susceptibility to a large range of diseases including dengue [22,23,24,25].

YF vaccines are effective and safe in most cases; however, some adverse events (AEs) and a few rare serious adverse events (SAEs) have been reported [26]. SAEs may occur following vaccination even in individuals that produce high neutralizing antibody levels [27,28,29], leading to the hypothesis of inborn innate response failure [30]. Thus, a few genetic studies have been carried out in isolated patients, revealing diverse backgrounds associated with a SAE outcome. These included a premature stop codon in the gene encoding the IFN-α/β receptor α chain (IFNAR1) [28], and pathology-associated SNPs in CCR5, RANTES [31], OAS1 and OAS2 [32], which encode a chemokine receptor, a chemokine ligand, and 2′-5′-oligoadenylate synthetase 1 and 2, respectively. However, to date, there are no studies associating the genetic background with an effective response after YF vaccination.

The C-type lectin domain containing 5A (CLEC5A) is an immune receptor abundantly expressed in myeloid lineages, which bind a variety of bacterial and viral pathogens eliciting proinflammatory responses [33]. The role of CLEC5A has been demonstrated for several viral infections, such as those caused by other flaviviruses, Zika, Japanese encephalitis, and dengue, as well as influenza [33,34,35,36]. Both human and mouse CLEC5A have been reported to bind to the dengue virus, and this receptor has already been related to severe dengue fever [37]. CLEC5A activation by flaviviruses and influenza virus induces phosphorylation of DNAX adaptor protein 12 (DAP12), triggering an inflammatory response characterized by production of IFNs, proinflammatory cytokines, and neutrophils extracellular traps (NETs) [37,38]. Blockade of CLEC5A with monoclonal antibodies attenuates inflammatory reactions, without downregulating IFN production, thereby protecting the host from virus-induced inflammatory reactions [39].

Moreover, the relationship between certain CLEC5A genetic variants and the expression levels of CLEC5A has been indicated in previous clinical studies [39]. Interestingly, CLEC5A SNPs have already been associated with the disease outcome for dengue [40].

Herein, we investigated the early expression of genes related to the innate antiviral response, the cellular immune phenotypes and the humoral response after immunization with the YF vaccine (YF17DD of Brazil). Moreover, we genotyped SNPs of the IFNG and CLEC5A genes in volunteers immunized with YF17DD and associated them to an effective humoral and early innate response, confirming the role of IFN-γ as a hub in YF vaccine protection and introducing CLEC5A as a new player in the YF vaccine immunologic response. Therefore, this study contributes to the understanding of the genotypic and phenotypic mechanisms associated with an effective immune response after YF vaccination.

## 2. Materials and Methods

### 2.1. Study Group and Blood Samples

Thirty-eight healthy volunteers, 23 women and 15 men, between 18 and 55 years old, were included in this study. In our sample, fifteen volunteers (11 woman and 4 men, mean = 33 year-old) presented no history of YF vaccination and twenty-three (12 woman and 11 men, mean = 36 years old) presented one previous YF17DD vaccination. All individuals were subcutaneously injected with a single 0.5 mL dose of the 17DD vaccine (Lot 007VFA010Z) as recommended by the manufacturer (Bio-Manguinhos, FIOCRUZ, Rio de Janeiro, Brazil). The volunteers were advised to report any clinical symptoms and adverse events after vaccination, but none were reported. The blood was collected before vaccination (day 0) and 3, 4, 5, 7, 10, and 60 days after vaccination (DAV). The study was conducted in accordance with fundamental ethical principles of the Declaration of Helsinki [41] and the Brazilian National Health Council on research involving human beings [42]. The study protocol was approved by the Research Ethics Committee of the National School of Public Health FIOCRUZ (protocol 145/01) and all volunteers gave their informed written consent before vaccination.

### 2.2. Purification and Cryopreservation of Human Peripheral Blood Mononuclear Cells (PMBCs) and Sera

PBMCs were obtained from 30 mL of heparinized venous blood. Blood samples were diluted 1:1 with RPMI 1640 medium (Sigma, St. Louis, MO, USA) and PBMCs were separated by performing a Histopaque™ gradient (d = 1077 g/mL, Sigma-Aldrich, San Luis, MO, USA) following manufacturer’s recommendations. The viability of the PBMCs was greater than 95%, as assessed by Trypan blue (Invitrogen, Carlsbad, CA, USA) exclusion. Approximately 10^7^ PBMCs were resuspended in 1 mL freezing solution [90% inactivated fetal bovine serum (FBS) (Gibco, Waltham, MA, USA) plus 10% DMSO (Sigma)] and stored in liquid nitrogen until use. For sera samples, tubes containing 8 mL of venous blood and serum separator clot (Beckman-Coulter, Brea, CA, USA) were centrifuged at 400× *g* for 10 min, and the supernatant was collected. Sera were stored at −20 °C until use.

### 2.3. Quantification of Antibodies against the Yellow Fever Virus

A new immunoenzymatic test, ViBI, was developed based on our previous work regarding the detection of neutralizing antibodies against tetanus and diphtheria toxins [43]. Plates were coated with 100 µL of 1 μg/mL 2D12 monoclonal antibody (Bio-Rad, Hercules, CA, USA) in carbonate-bicarbonate buffer pH 9.6 and incubated overnight at 4 °C. The plates were washed 5 times with PBS/T washing buffer (PBS pH 7.4 with 0.05% Tween-20) and subsequently blocked with blocking/diluent solution (BDS) at 100 μL/well [PBS/T, 0.05% bovine serum albumin (BSA), 3% FBS and 5% skimmed milk] for 1 h at 37 °C. Sera from immunized volunteers at 60 DAV were diluted 1:10 in BDS to perform serial twofold dilutions. For the standard curve, twofold serial dilutions of the monkey anti-yellow fever antibody serum (National Institute for Biological Standards and Control, UK), in BDS, ranging from 1000 to 0.008 mUI/mL plus 20 µg/mL were performed. A virus dose control (virus plus BDS only) and a blank control (BDS only) were also included. After 1 h at room temperature, rocking at 300 rpm, the plate was washed with PBS/T and incubated with 100 μL/well of the 2D12 HRP-conjugated antibody diluted 1:800 in BDS and incubated for 1 h at 37 °C. After washing, 100 μL/well of TMB plus (KemEnTec, Taastrup, Denmark) was added for 15 min, followed by 100 µL/well of 2M H_2_SO_4_. Endpoint measurements were made at 450 nm using a VersaMax (Molecular Devices, San José, CA, USA). Calculations were performed with SoftMax Pro V5.4 (Molecular Devices, San José, CA, USA)and Microsoft Office 365Excel (Microsoft, Albuquerque, NM, USA).

### 2.4. Immunolabeling for Flow Cytometry

PBMCs from 0, 3, 5, and 7 DAV were thawed in RPMI medium (Thermo Fisher, Waltham, MA, USA) and 2 × 10^5^ live cells were added to 96-well plates with 0.02 MOI of YF17DD and maintained for 24 h with RPMI 1640 medium in a moist chamber at 37 °C in 5% CO_2_ atmosphere. For positive control, cells were incubated with 0.01 µg/tube of phorbol 12-myristate 13-acetate (PMA) and 0.2 µg/tube of ionomycin (Sigma-Aldrich). As a negative control, cells were incubated with RPMI 1640 medium (mock).

After incubation, cells were washed in PBS pH 7.2 supplemented with 1% BSA and 0.1% sodium azide (NaN_3_), then immunolabelled with the monoclonal antibodies anti-CLEC5A-PE, and phenotypical markers of monocytic lineage (anti-CD14-APC, and anti-CD16-FITC), T cells (anti-CD3-APC-Cy7), CD4^+^ T cells (anti-CD4-BV421), cellular activation (anti-HLA-DR-PE-Cy5), and apoptosis (anti-CD95-PE-Cy7) (Beckman–Coulter), for 30 min at 4 °C, according to manufacturer’s recommendation. Cytometric analyses were carried out in a BD LSR Fortessa ^TM^ Cell Analyzer (Becton Dickinson, Franklin Lakes, NJ, USA). For each sample, 10,000–20,000 events were acquired, and analyses were made with the FlowJo V10 software (FlowJo LLC, Ashland, OR, USA). The CD8^+^ T cells were characterized by excluding CD4^+^ from CD3^+^ cells. The gating strategy shown in Appendix A.

### 2.5. DNA and RNA Extraction

Genomic DNA extraction was performed from thawed cryopreserved PBMCs (10^7^ cells) using the salting out methodology [28] and DNA was resuspended in TE buffer (5 mM Tris-HCl, 0.1 mM EDTA). The RNA was extracted using TRIzol ^TM^ reagent (Thermo Fisher), followed by complementary DNA (cDNA) production from 250 ng of total RNA using the High-Capacity cDNA Reverse Transcription Kit (Thermo Fisher), both procedures according to the manufacturer’s instructions. After isolation, the nucleic acids were quantified in a spectrophotometer (Nanodrop Technologies, Wilmington, DE, USA) and stored at −20 °C until the time of use.

### 2.6. Expression of Genes Related to an Antiviral Response

The cDNA from each sample was used to quantify the mRNA of CLEC5A, DAP12, STAT1, STAT2, IRF7, IRF9, OAS1, RNASEL, IL6, IL12, CXCL10, NOS1, AIM2, IFI16, IFNGR1 and IFNG, as well as the reference genes GAPDH and RPL13 (genes and primer descriptions are given in Appendix A). To this end, Sybr Green master mix (Thermo Fisher) with 200 nM of each primer (forward and reverse) and 10 ng of each cDNA were used in a final reaction volume of 10 µL. The standard cycling conditions used were a pre-dwell for 10 min at 95 °C; 15 s at 95 °C, 4 min at 60 °C, for 45 cycles. Gene expression was carried out using the ViiA™ 7 Real-Time PCR System (Thermo Fisher). Data obtained were normalized to the average cycling threshold value of the reference gene RPL13, and then the difference in normalized cycling threshold values between 4, 7, and 10 DAV versus 0 DAV was calculated, generating 2ΔΔCt. Fold change (2ΔΔCt) was compared between the DAV times by a Kruskall–Wallis test with Dunn’s post-test using the GraphPad Prism 5 software.

### 2.7. Genotyping and Genetic Analysis

The DNAs of immunized volunteers were genotyped for SNPs using Taqman probes. For CLEC5A, two SNPs were genotyped: rs13237944, (+7569 C > A) and rs1285933 (C_9506735_10). As for IFNG, three SNPs were genotyped: rs2430561 (AH20TEB), rs2069718 (C_15799728_10), and rs1861493 (C_2683476_10) (SNPs descriptions are given in Appendix A). Genotyping was performed by allelic discrimination real-time PCR performed in a StepOne Plus instrument (Life Technologies, Carlsbad, CA, USA). Reactions containing 20–40 ng of DNA in a final volume of 20 µL including 10 µL of TaqMan Genotypin Master Mix (Life Technologies) and 0.5 µL of each TaqMan probe were used in standard cycling conditions: a pre-dwell for 5 min at 95 °C; 15 s at 95 °C, 30 s at 60 °C, for 40 cycles.

## 3. Results

### 3.1. Activation of CLEC5A^+^ Monocytes and T Lymphocytes after YF17DD Vaccination

After in vitro YF17DD stimulation of peripheral blood mononuclear cells collected from vaccinated volunteers, cellular activation was assessed through the expression of HLA-DR^+^ (Appendix A). It was observed that the frequency of the subpopulation of activated monocytes not expressing CLEC5A did not exhibit statistically significant modulations over time (Figure 1A), unlike activated monocytes (CD3^−^CD95^−^CD14^+^HLA-DR^+^) expressing CLEC5A, which were significantly increased on the fifth day after vaccination (Figure 1B,C). These results suggest that CLEC5A is involved with monocyte activation after YF exposition and could act in the initial YF interaction with cells. Activated CD4^+^ T cells (CD4^+^HLA-DR^+^) and CD8^+^ T cells (CD8^+^HLA-DR^+^) exhibited a gradual increase over time, with a peak at 7 DAV (Figure 1C–E) and a positive correlation considering all times analyzed (R = 0.71; *p =* 0.00001). In summary, the kinetics of YF17DD response involved a fist line response of CLEC5A^+^ activated monocytes at 5DAV, followed by an activation of CD4^+^ and CD8^+^ cells 7 DAV (Figure 1F).

### 3.2. Expression of CLEC5A and Genes Related to Interferon Pathways Has a Positive Correlation with YF17DD Vaccine Immunogenicity

Relative quantification of mRNA in the PBMCs of 17DD first-time vaccinated individuals (*n* = 15) demonstrated that genes related to the antiviral and inflammatory responses have an important role in the early events after YF vaccination. The expression of CLEC5A varied according to time, with high levels at 4 DAV and 10 DAV (Figure 2A), presenting a significant positive correlation with the anti-YF antibody production at 10 DAV (Figure 2C). Furthermore, expression of CLEC5A did not show any correlation with DAP12 (Figure 2B,C), a gene that encodes the CLEC5A associated protein. At 7 DAV, there was a positive correlation of CLEC5A expression with the proinflammatory cytokine IL6 and the chemokine CXCL10, and genes related to the innate antiviral response (STAT1, IRF7, IRF9, OAS1, and RNASEL) (Figure 2B), suggesting that CLEC5A contributes to a proinflammatory and antiviral environment.

Interestingly, genes related to type II IFN pathway presented an early interaction with other antiviral genes, since times 7 DAV. At 7 DAV, an increase in the relative expression for the majority of analyzed genes was observed, which was significant for CXCL10 (*p* < 0.0001) (Figure 2A), a downstream chemokine of type I and II IFN pathways. Besides, at 7 DAV, expression of AIM2, IFNGR1 and IFNG presented a positive correlation with ViBI antibody levels (Figure 2B). Data also suggest at 10 DAV an interaction network related with humoral response after YF vaccination: AIM2, IFI16, and STAT1, typical products of the IFN-γ pathway, presented a positive correlation with CLEC5A, STAT2, IRF7, IRF9, OAS1, RNASEL and the antibody levels (Figure 2C).

### 3.3. Polymorphism in CLEC5A and IFNG Genes Modulates the Immunologic Response to the YF17DD Vaccine

In order to investigate whether SNPs in the CLEC5A and IFNG genes are related to the YF17DD vaccination immunologic response, the DNA of first-time vaccinated and revaccinated individuals (*n* = 38) were genotyped for three SNPs of the IFNG gene, rs2430561, rs2069718, and rs1861493, and two SNPs of the CLEC5A gene, rs13237944 and rs1285933. It was demonstrated that at 7 DAV a significant upregulation of CLEC5A expression occurred in individuals with genotypes AC/AA rs13237944 (Figure 3A), and of IFNG in individuals with genotypes AG/AA rs2430561 and AT/AA rs2069718, when compared with the the other possible genotypes (Figure 3B,C). Protein levels of CLEC5A in ex vivo-activated monocytes and circulating IFN-γ did not demonstrate significant differences when clustering by genotypes (Appendix A). Regarding anti-YF antibody production, data only showed a trend towards a higher antibody production at 60 DAV in genotypes marked by high expression of CLEC5A and IFNG (Figure 3D–F), although more sampling efforts are necessary to definitively associate the SNPs analyzed with the anti-YF antibody production after YF vaccination.

## 4. Discussion

The yellow fever vaccine has been used for more than 70 years on more than 400 million people, with a remarkable record of safety and efficacy. Despite this, there have been several recent outbreaks highlighting the continuous necessity of YF vaccination. Angola and the Democratic Republic of Congo experienced large outbreaks in 2015 and 2016, followed by Brazil and Nigeria in 2017 and 2018. Brazil has experienced increased YF outbreaks due to an ongoing YF epizootic transmission that has expanded from endemic zones to areas near the megacities of Rio de Janeiro and São Paulo [44]. In less than 18 months, 1833 confirmed cases and 578 deaths were recorded, most of them reported in the Rio de Janeiro–São Paulo axis [45]. Outbreaks are accompanied by massive vaccination schemes where rare SAE cases are eventually detected [45,46]. Furthermore, unvaccinated travelers visiting Brazilian endemic areas have acquired and died from YF with a higher frequency, compared with past decades, making YF vaccination a worldwide demand [47].

Here we investigated the integration between the innate and adaptive immune responses following YF17DD vaccination. Induction of innate virus-sensing genes and interferon production was observed, orchestrated by major transcription factors, including STAT2 and IRF7. Effector cells of the immune system, including early CD8^+^ T cell responses, a hallmark of YF17DD efficacy, were detected [4,48]. Herein, in the light of human genetic background, we demonstrated for the first time a possible role for the CLEC5A receptor after YF vaccination, and moreover, we focused on the importance of IFN-γ as a key factor for vaccine immune response development.

First, it was demonstrated that a balanced proinflammatory environment is necessary for neutralizing antibody production in YF vaccination [49]. In this context, CLEC5A has been shown to be an important pattern recognition receptor for flavivirus and is involved with cytokine release from monocytic cells [37]. As shown before, the blockade of CLEC5A^−^ dengue virus interactions could attenuate inflammation and maintain host immunity so as to clear virus, in an IFN-independent manner [37]. Our results revealed that high gene expression and abundance of CLEC5A on the surface of activated monocytes are correlated with YF vaccine immunogenicity, as measured by antibodies at 60 DAV. Additionally, data demonstrated an early positive correlation between the expression of CLEC5A and the proinflammatory factors IL6 and CXCL10, already described as T cell activators [50,51,52]. Moreover, activated CD4^+^ and CD8^+^ T cells were also positive correlated with CLEC5^+^ monocytes. Based on these results and the role of CLEC5A described for other flaviviruses infections, we hypothesize that CLEC5A could act as a YFV receptor with a downstream activation cascade that stimulates proinflammatory cytokines secretion, helping the activation of T cells [33,36,37,38]. Nevertheless, we could not observe a correlation between levels of gene expression of CLEC5A and DAP12, the protein that is phosphorylated downstream CLEC5A activation [37,38]. It could be due to gene expression approach that does not cover phosphorylation activities, which probably is occurring in this case. At the genetic level, it was reported that individuals carrying the rs1285933 CC genotype for CLEC5A present high levels of CLEC5A mRNA and milder dengue infections compared to those with other genotypes in the Brazilian population [53]. Despite a low sample size, according to what was observed for dengue, here we show that individuals with genotypes AA/CA for another SNP as eQTL (expression quantitative trait loci) for YFV (rs13237944), which is associated with high levels of CLEC5A mRNA.

Previous investigations on the cellular response following YF vaccination have reported the activation of CD4^+^ T cells by day seven, CD19^+^ T cells by day 15, and CD8^+^ T cells by day 30 after vaccination [30,54]. ELISpot has been used to show that IFN-γ was significantly increased on day 15 after YF17DD immunization [55]. Our data depicted events of CD8^+^ T cell activation as early as seven days after vaccination. We also covered the importance of innate network in response to YF17DD up to seven days after vaccination, presenting the early correlations between type II IFN genes and the transcription factors IRF7, IRF9, STAT1, and STAT2, as well as OAS1. Together, these data suggest that the IFN-γ response produced by CD8^+^ cells may start seven days after YF17DD vaccination, evidencing its prolonged and crucial role for YF17DD immunogenicity, according to similar studies [4,36]. Another group has observed similar results in humans [21]. Markedly, our results corroborated this hypothesis, as expression of genes related to the IFN-γ pathway (AIM2, IFI16, and STAT1) at 10 DAV displayed a high correlation with anti-YF antibody levels. Despite low sample size, individuals with the genotypes AT/AA of rs2430561 and AG/AA of rs2069718 showed greater IFNG expression levels, highlighting these eQTL loci as possible targets for future association studies on YF immunogenicity.

In conclusion, the present study showed that the early events elicited after YF17DD vaccination are, at least in part, associated with and controlled by human genetic background, as observed here at CLEC5A and IFNG. Nevertheless, it is likely that several other SNPs and environmental and biological factors also contribute to the complex immune response triggered after YF17DD vaccination, in a way that we cannot extrapolate that long-lasting immunity is due to host genetics. Moreover, we confirmed that the IFN-γ pathway acts as a hub in YF17DD protection and as well we introduced CLEC5A as a new player during the YF response, at both the phenotypic and genotypic levels.

## Figures and Tables

**Figure 1 viruses-13-00096-f001:**
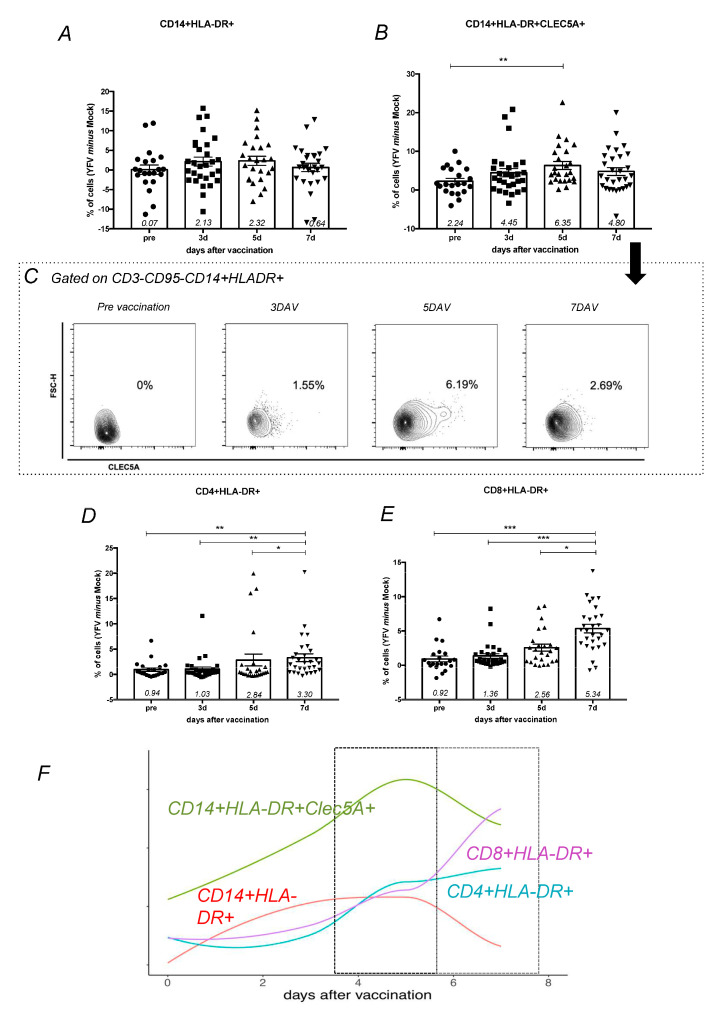
Phenotypes of monocytes and T lymphocytes during vaccination follow-up. Peripheral blood mononuclear cells (PBMCs) from vaccinated volunteers (*n* = 38, 15 first time vaccinated and 23 s time vaccinated) were stimulated for 24 h with YF17DD in vitro. Bar-graphs show the cell population frequencies of (**A**) activated monocytes, represented as CD3^−^CD95^−^CD14^+^HLA-DR^+^, and (**B**) activated monocytes expressing CLEC5A, represented as CD3^−^CD95^−^CD14^+^HLA-DR^+^CLEC5A^+^. Cell population frequencies were calculated by frequency in YF17DD stimulated experimental condition minus frequency in unstimulated (mock) experimental condition. (**C**) Cells from one subject are represented as a density plot (FlowJo Tree Star^®^) showing the kinetics of CLEC5A expression on activated monocytes. (**D**) Activated CD4^+^ T cells are represented as CD3^+^CD95^−^CD4^+^HLA-DR^+^. (**E**) Activated CD8^+^ T cells are represented as CD3^+^CD95^−^CD4^−^HLA-DR^+^. Cell population frequencies were compared between time points using Kruskal–Wallis with Dunns post-test, with significant *p* value represented as * *p* < 0.05, ** *p* < 0.01, *** *p* < 0.001. Each point corresponds to one individual analyzed, with median and standard error of groups. (**F**) Graphic representation of cellular phenotypes during vaccination follow-up. Lines represent these percentage difference distributions over time for CD14^+^HLA-DR^+^CLEC5A^+^ (green), CD14^+^HLA-DR^+^ (red), CD8^+^HLA-DR^+^ (purple), and CD4^+^HLA-DR^+^ (turquoise).

**Figure 2 viruses-13-00096-f002:**
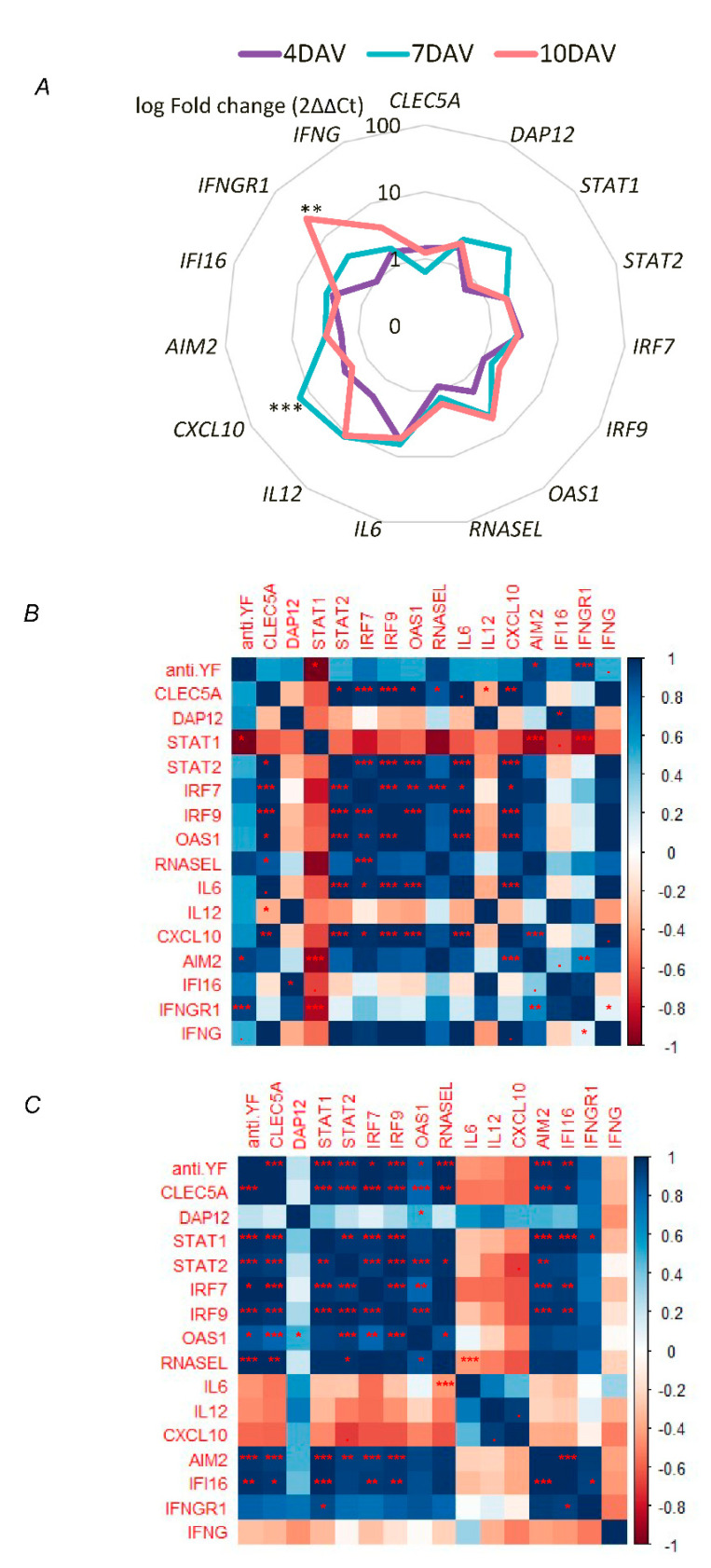
Gene expression profile of the early responses in first time YF17DD-vaccinated volunteers (*n* = 15). (**A**) Radar chart representing the average log10 fold change (2ΔΔCt) of each gene and day after vaccination (DAV). Mean 2ΔΔCt values of CLEC5A, DAP12, STAT1, STAT2, IRF7, IRF9, OAS1, RNASEL, IL6, IL12, CXCL10, NOS1, AIM2, IFI16, IFNGR1, and IFNG clustered by 4 DAV (red), 7 DAV (yellow), and 10 DAV (blue). Fold change (2ΔΔCt) was compared between DAV by Kruskall–Wallis test with Dunn’s post-test using the GraphPad Prism 5 software. The heatmaps represent the correlation R values according to the Z score for 2ΔΔCt values of genes analyzed and anti-YF antibody production at (**B**) 7 DAV, and (**C**) 10 DAV. Spearman correlation and heatmaps by library “corrplor” from R-project. *p* value is represented as *p* < 0.1, * *p* < 0.05, ** *p* < 0.01, *** *p* < 0.001.

**Figure 3 viruses-13-00096-f003:**
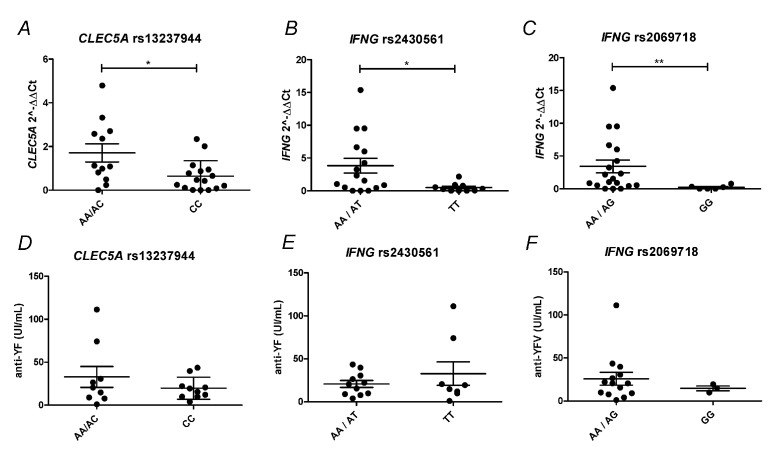
CLEC5A and IFNG SNPs associated with YF17DD immunologic responses. (**A**–**C**) Relative expression of CLEC5A and IFNG at 7 DAV according to SNP genotyping. The expression of CLEC5A and IFNG was stratified according to (**A**) rs13237944, (**B**) rs2430561, and (**C**) rs2069718 genotypes (n = 38). (**D**–**F**) Anti-YF antibody production according to SNP genotyping. Anti-YF antibody production at 60 DAV was stratified according to (**A**) rs13237944, (**B**) rs2430561, and (**C**) rs2069718 genotypes (*n* = 19). Each dot corresponds to one individual analyzed, with median and standard error of groups. Statistical analysis was determined using Mann–Whitney test. *p* value is represented as * *p* < 0.05, ** *p* < 0.01. DAV: days after vaccination.

## Data Availability

Data available on request due to restrictions eg privacy or ethical. The data presented in this study are available on request from the corresponding author. The data are not publicly available due to ethical reasons.

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
