# Peer review of "Activation of an Effective Immune Response after Yellow Fever Vaccination Is Associated with the Genetic Background and Early Response of IFN-γ and CLEC5A"

_viruses, 2021, doi:10.3390/v13010096_

Round 1

Reviewer 1 Report

This is a good experimental and clinical paper. But I have some concerrns

1) Minor: While the authors mentioned how Theiler developed the first YFV vaccine, they did not explain exactly how YFD17DD was developed. Is YFD17DD identical to the vaccine that Theiler developed? If not, how different is it? This needs to be elaborated

2) Major: This paper is an easier read for someone with a strong immunology backgournd but may be a more difficult read for those without a strong immunology background. The fact that this journal is "Viruses" basically means that the readers are likely to be from various background with interest in virology. They could have background in areas such as biochemistry and not necessarily in immunology. I think the authors could make it easier to the readers in general. One way to do it is by making a summary table of the genes/proteins involved. The table will include columns for abbreviations, name and a sentence or two describing the gene/protein. For example, the first item in the table will be CLEC5A, "C-Type Lectin Domain Containing 5A", "immune receptor abundantly expressed in myeloid lineages, which bind a 68 variety of bacterial and viral pathogens eliciting proinflammatory responses "..etc.

The authors therefore need to make the paper more interesting and readable for the readers at Viruses.

Reviewer 2 Report

see attached

Reviewer 3 Report

In this manuscript the authors (Azamor et al.) present their studies of potential immunogenetic influences on the vaccine response to the yellow fever vaccine, YF17DD, with a special focus on interferon-gamma (IFNg) and the monocyte expressed molecule C-type lectin domain containing 5A (CLEC5A), which has previously been shown to play a role in the immune response and pathogenesis of dengue. While the approach appears appropriate and the analysis appears thorough, the study is very limited in scope with regards to participant numbers and hence statistical power. Moreover, for the analysis of the effect of polymorphism in the CLEC5A and IFNG genes, they lump together the two groups of participants, i.e., first time vaccinated and booster vaccinated. For all other parameters there is no non-vaccinated control group. While they do mention that the participants were between 18 and 55 years of age, there is no information whether the age distribution was even between the two groups, nor is there any information about gender.

It is unclear how the authors can conclude – in lines 183-185 – the expression of CLEC5A on activated monocytes “could acts [sic] in the initial YF interaction with cells”. Is this based on findings in the dengue virus system? Finally, the antibody response was only followed to day 60 post vaccination, so to conclude that “early events elicited after YF17DD vaccination, determined by human genetic background, are key to developing a long-lasting immune response”, is clearly an overstatement if not disingenuous. This conclusion should be toned down unless the authors can clearly show long-lasting immunity in these participants.

The manuscript could be further improved by addressing the following:

  1. The authors introduce the abbreviation DAV (days after vaccination) already in the abstract, but several of the figures are labeled “days post vaccination” or DPV, e.g., Figure 1C and Figure 2A, and in a few places in the text they also say ‘days post vaccination’. While I personally prefer the latter expression, the authors should at least be consistent between figures and text.
  2. Line 51: correct to CD8+ T cell response
  3. Line 68: the abbreviation CLEC5A should be defined here – and a sentence should in any circumstance preferentially not start with an abbreviation.
  4. Throughout the manuscript there are multiple examples of missing spaces between words or between a number and a word (e.g., line 93, multiple examples of 7DAV, 10DAV etc.). And in lines 124 and 126 it says “1 hat” instead of “1 h at”This should be corrected.
  5. Line 106: correct to Histopaque.
  6. Line 130: change to “Immunolabeling for flow cytometry”. The cells are not stained in the true sense of the word, they are reacted with antibodies, hence labeled. The wording should also be changed accordingly in line 138.
  7. Line 150: correct to ‘according to the manufacturer’s”
  8. Lines 213 & 216: change to “related to”.
  9. Line 241: what is meant by “the otherwise genotypes”? Should it read “the other genotypes”?
  10. Line 242: correct to “did not”
  11. The statements in lines 301-303 are unclear and should be rewritten.
  12. Line 309: correct to “on monocytes”

END

Round 2

Reviewer 1 Report

The authors need to be careful of grammar:

"Hence, the early events after YF vaccination consists in i) a complex 64 modulation of innate immune.." is not grammatically correct.

Reviewer 2 Report

no more concerns

Author Response

Dear reviewer, 

We would like to thank you for your great contributions.

Best Regards,
Tamiris Azamor

Reviewer 3 Report

The authors are commended for addressing most of this reviewer's concerns and suggestions. However, I remain concerned that too much is made of this considering the small cohort studied, and that the authors rely on historical data for vaccine response longevity and do not actually assess that in the study cohort. If the YFV vaccine is as good as claimed, then perhaps these gene associations are only coincidental and not of actual importance for the longterm response?
